# Impact of housing allowance programme on the physical and mental health of households in South Korea

**Saehim Kim** [ID]**, Saebae Ryu, Kyuhyun Park, Myeong-Hun Lee** [ID]*

Graduate School of Urban Studies, Hanyang University, Seoul, South Korea

* mhlee99@hanyang.ac.kr

## Abstract

Excessive housing costs significantly affect household financial stability and overall well-being. This study investigated the impact of South Korea's housing allowance programme on the physical and mental health of household heads, utilising data from the Korea Welfare Panel Study (2009–2021). To overcome selection bias, we employed propensity score matching to construct a comparable control group. We then estimated a two-way fixed effects event study model to assess the dynamic health impacts following the programme's significant reform in 2015. Our analysis confirms the absence of pre-existing differential trends, supporting the validity of our research design. The results indicate that the policy's positive effects were not immediate but emerged over time. A statistically significant reduction in depression appeared approximately four years post-reform. For physical health, a consistent and statistically significant improvement was observed from 2017 onwards, highlighting a delayed but sustained positive impact. The findings, validated by a rigorous quasi-experimental design, emphasise the critical role of housing welfare policies in promoting health equity and suggest the benefits of such policies may accumulate over time.

## 1. Introduction

Excessive housing cost burden is a major factor contributing to household financial difficulties and is a key element of housing deprivation. Unaffordable housing costs compel households to opt for lower-quality housing, negatively impacting household well-being [1]. Poor residential environments are also known to affect the health of residents adversely. Previous studies have demonstrated that changes in residential areas or reductions in living space negatively impact residents' mental health, and unfavourable environments, such as basement dwellings, harm residents' physical health [2]. These findings suggest that alleviating the housing cost burden is crucial for providing economic stability and promoting overall resident health. Consequently,

**Data availability statement:** We have uploaded the minimal dataset necessary to replicate our findings to Figshare. The dataset is now publicly available and can be accessed via the following DOI: [10.6084/m9.figshare.29938802].

**Funding:** The author(s) received no specific funding for this work.

**Competing interests:** The authors have declared that no competing interests exist.

policies aimed at addressing housing deprivation can be considered significant factors positively impacting society as a whole [3].

Governments in many countries operate housing assistance programmes to mitigate the housing cost burden for low-income households. Recently, there has been a trend towards adopting demand-side housing policies, such as housing allowances, rather than supply-side policies, such as public rental housing [4]. Housing allowances typically subsidise the rent for residents renting their homes. These allowances are vital in helping recipients avoid housing deprivation and reducing the economic and health challenges associated with a high housing cost burden. Thus, mitigating the housing cost burden through housing allowances can positively impact recipients' overall quality of life and health, going beyond mere financial assistance [5,6]. It is crucial to understand the overall impact in order to design and implement housing welfare policies effectively.

South Korea also operates a housing allowance programme to support low-income renter households with housing cost burdens. It operates as a direct cash transfer, providing a monthly subsidy to eligible low-income renter households to help them afford housing in the private market. This tenant-based approach differs from the more diverse system in the U.S., which includes not only tenant-based vouchers but also project-based assistance where subsidies are tied to specific buildings [7]. This programme underwent significant reform in 2015, greatly expanding both the scope of beneficiaries and the level of support provided [4]. Key changes included raising the income eligibility threshold from 33% to 43% of the median income and basing the subsidy amount on a standardized regional rent, leading to an increase in both the number of beneficiaries and the average payment they received [4,8]. These changes aimed to extend benefits to more households experiencing housing instability. This major policy expansion provides a valuable opportunity to examine the causal impact of a large-scale housing allowance program on the well-being of recipients.

The primary goal of the housing allowance is to help economically vulnerable households secure a more stable and adequate residential environment. The expansion of this programme aims to provide economic stability to households burdened by high housing costs, thereby improving overall social welfare and quality of life for individuals. Several studies have examined the impact of the housing allowance programme on alleviating housing cost burden for households and securing economic stability. However, there is still insufficient empirical evidence to demonstrate the significant impact of the housing allowance system on the physical and mental health of individuals by helping low-income households escape housing deprivation.

South Korea's housing allowance programme was reformed in 2015 and has operated ever since. Thus, there are insufficient empirical research findings on the effects of the policy, highlighting the need for a multifaceted assessment of its impacts. This study aims to empirically analyse the impact of South Korea's housing allowance programme on the physical and mental health of household heads. To provide a rigorous causal assessment, we employ propensity score matching (PSM) to construct a comparable control group and a two-way fixed effects (TWFE) event study model to

analyse the dynamic effects of the policy changes after 2015. By examining the intersection of housing welfare and health through this robust quasi-experimental design, this study is expected to contribute to academic discussions and provide credible insights for policymakers who design and implement housing welfare policies.

## 2. Literature review

### 2.1. Purpose and role of the housing allowance programme

This section introduces the necessity and purpose of housing allowances, compares the programme in South Korea before and after its reform, and discusses the impact of policy changes on alleviating the housing cost burden.

Housing cost burden significantly affects the quality of life for individuals and households. Various countries implement housing allowance programmes to mitigate this burden for low-income households. Examples include the Housing Choice Voucher in the United States, housing benefits in the United Kingdom, and the housing allowance programme in the Netherlands [9]. Despite differences in form, these programmes share the common goal of providing affordable, improved residential environments for low-income households. They offer greater freedom of housing choice by subsidising housing costs, enabling better living conditions.

While these programs share the common goal of alleviating housing cost burdens for low-income households, their structures and delivery mechanisms differ significantly based on each country's unique policy landscape. The U.S. system, for instance, is characterized by a diverse range of programs, including project-based Public Housing and tenant-based Section 8 vouchers, which are administered in a complex, multi-layered system [7]. The U.K.'s approach has been shaped by extensive welfare reforms, with a strong emphasis on income-related housing benefits that vary regionally due to devolution [10,11]. In contrast, South Korea's housing allowance operates as a direct cash transfer, forming a key pillar of its anti-poverty strategy within an evolving welfare state [12]. This distinction is important; while the Korean program offers recipients greater autonomy in the private market, it differs from systems that also incorporate supply-side incentives or place-based social housing.

The South Korean government operates a housing allowance programme to reduce housing costs. Initially, the livelihood protection system provided general protection from the enactment of the National Assistance Act in 1961 until the 1990s. In 1999, the National Basic Living Security Act replaced the National Assistance Act, leading to the implementation of the National Basic Livelihood Security Programme in 2000 [13]. During the National Assistance Act years, housing costs were part of the livelihood allowance. Post-Act, the government separated housing allowances to offer recipients better residential environments. Initially included in the integrated allowance of the National Basic Livelihood Security Programme, housing allowances were separated following the transition to a customised allowance system in July 2015 [8].

Before the reform, renter households meeting criteria for obligatory providers and with recognised incomes below the minimum cost of living (33% of the standard median income) received 22.032% of the amount calculated by subtracting the recognised income from the standard benefit amount (approximately 80% of the minimum cost of living) as a housing allowance [14]. Post-reform, the range and criteria for obligatory providers' support were relaxed, and the recognised income standard was raised to 43% of the median income provided [8]. The rental allowance is now based on the standard rent for each region, with higher standard rents for areas with high rental fees, such as the capital area [15]. The standard rent is established to ensure that citizens can live in housing that meets minimum residential standards. Calculated by the Ministry of Land, Infrastructure and Transport using Korea Housing Survey data, this rate is applied differentially across four national zones based on the number of household members. The reformed policy determines the housing allowance level by considering factors such as the recognised income of the recipient household, the number of household members, the type of residence, and the housing cost burden [14].

Empirical studies have examined the effects of the housing allowance programme, concluding that it effectively alleviated the housing cost burden for those with housing vulnerability [16]. In South Korea, the programme also alleviated the

housing cost burden for households, especially after the reform, which increased the number of recipients and the amount paid [4].

## 2.2. Housing affordability and individual health

This section discusses the negative impacts of excessive housing cost burden on households. We review previous studies to understand how excessive housing expenditures negatively impact individuals and households economically and in terms of financial stability and health.

To explain the impact of housing insecurity on health, the stress process model offers a valuable theoretical framework. Recently, [17] proposed a conceptual model viewing housing insecurity as a chronic stressor that leads to physiological and epigenetic health issues through several pathways, including health behaviors, psychosocial resources, and structural resources. According to this model, stressors such as the housing cost burden extend beyond mere financial hardship; they can lead to physical health deterioration by fostering unhealthy lifestyles or undermining mental stability. Grounded in this theoretical background, our study investigates the hypothesis that South Korea's housing allowance program can positively influence the mental and physical health of recipient households by mitigating these stress pathways.

As a key component of this stress process, an excessive housing cost burden—generally defined as exceeding 30% of household income [18,19]—deteriorates the economic stability of households and negatively impacts overall life quality [1,20]. Excessive housing expenditures reduce spending on other areas related to quality of life [12,21], adversely affecting the health of individuals and household members.

Previous studies have linked excessive housing cost burden to deteriorating physical health. A significant correlation exists between increased housing expenditures and preventable and treatable mortality [22], which can be explained by the reduction in health-related expenditures due to excessive housing costs.

Moreover, higher housing cost burden has led to higher mortality rates from accidents and self-harm [23]. This indicates that housing cost overburden can harm the mental health of individuals and household members. In fact, stress caused by high housing costs leads to mental health problems [24,25]. Specifically, it has been found that the housing cost burden can lead to depression [26].

These adverse effects of housing cost burdens are more apparent and disproportionately concentrated among socially vulnerable groups. For instance, the burden often falls heavily on households headed by women [16], a vulnerability that is exacerbated in regions undergoing economic adjustments that increase housing costs [27]. Similarly, older adults find it more difficult to escape excessive housing cost burdens, which significantly impacts their mental health [2]. This challenge for the elderly is also globally observed, as rising housing costs can make housing unaffordable for a significant portion of the aging population [27]. The housing affordability crisis also severely affects other vulnerable populations, such as individuals with disabilities who face an elevated risk of eviction [28], and young people who are pushed into less secure rental situations [29].

Furthermore, the economic consequences of unaffordable housing often lead to socio-spatial segregation, as low- and middle-income households are frequently displaced to areas with higher unemployment [30], while urban sprawl pushes them to peripheral areas, reinforcing inequality and reducing opportunities for upward mobility [31]. Ultimately, this burden translates into direct health consequences. In regions with severe housing affordability issues like Hong Kong, the deprivation associated with high housing costs is directly linked to deteriorations in both physical and mental health [32]. Overall, excessive housing cost burdens lead to negative economic impacts and deteriorate health, thereby lowering the overall quality of life [33]. This phenomenon is especially evident among socially vulnerable groups. As our study uses gender and age as key analytical variables to assess the health outcomes of Korea's housing allowance, this body of international literature provides a critical framework.

A substantial body of evidence indicates that governmental housing assistance programs can ameliorate these negative health effects by reducing housing insecurity. Research has shown that rental assistance is associated with significant

improvements in mental health, with recipients reporting less psychological distress compared to those on waitlists [5]. The benefits extend to physical health outcomes and the management of chronic conditions; for instance, housing assistance has been linked to a reduced likelihood of uncontrolled and undiagnosed diabetes [34,35]. A key mechanism for these improvements is the alleviation of material hardship, which allows households to redirect resources toward other health-promoting necessities. For example, studies have found that rental assistance improves food security and increases the consumption of fruits and vegetables [36]. This literature provides a strong empirical basis for hypothesizing that policies designed to reduce housing cost burdens can serve as effective public health interventions.

This study aims to empirically analyse the impact of South Korea's housing allowance programme on the physical and mental health of households. Each country's programme has different characteristics, and since South Korea reformed its own in 2015 provided [4], this study is significant in analysing the impact of the post-reform programme on household health. While previous studies primarily focused on the economic impact of housing allowance programmes, this study distinguishes itself by examining the impact on household health.

## 3. Methods

### 3.1. Data and methodology

This study analysed data from the Korea Welfare Panel Study, jointly conducted by the Korea Institute for Health and Social Affairs and the Seoul National University Institute of Social Welfare. The Korea Welfare Panel Study is an annual survey designed to investigate citizens' living conditions and welfare needs, evaluate policy effectiveness, and inform the development of new policies and systems in South Korea [37]. Our analysis uses an unbalanced panel spanning from 2009 to 2021 (excluding 2018), drawn from the Korea Welfare Panel Study. The initial full dataset comprised 10,516 unique households and a total of 72,513 household-year observations. We applied a pre-specified rule for outlier handling, winsorizing continuous variables at the 0.5th and 99.5th percentiles. After listwise deletion of cases with missing data on key variables, the final sample constructed for propensity score matching consisted of 3,486 unique households. This analytical sample is defined as follows: the treatment group (N = 302 households) comprises those that received the housing allowance, and the control group (N = 3,184 households) consists of never-recipients. The underlying allowance receipt status is a time-varying variable in our panel data. Crucially, as the 2015 reform expanded the eligibility criteria, we are able to observe households that begin receiving the allowance for the first time only after this expansion.

To estimate the causal impact of the housing allowance on the health of household heads, this study employs a difference-in-differences (DID) research design. A critical challenge in this observational setting is the potential for selection bias, as households receiving the allowance may systematically differ from non-recipients. To address this, we first use propensity score matching (PSM) to construct a comparable control group. By matching recipients to non-recipients based on a rich set of pre-reform observable characteristics, we ensure that our treatment and control groups are balanced, thus minimizing selection.

Using this matched sample, we then estimate a two-way fixed effects (TWFE) event study model. This approach is superior to a traditional DID model for two reasons. First, the two-way fixed effects control for both time-invariant individual characteristics (individual fixed effects) and common year-specific shocks (year fixed effects), which mitigates bias from unobserved heterogeneity [38–40]. Second, a TWFE event study replaces a single pre-post treatment indicator with a series of year-by-year interaction terms. This approach allows researchers to formally test the key parallel trends assumption and to trace the dynamic, year-over-year effects of the policy following the 2015 reform [41,42]. This model is estimated separately for each of our two health outcomes: depression and self-rated health.

Our analytical strategy relies on two key assumptions. First, the PSM relies on the conditional independence assumption, which we test by assessing the covariate balance between the matched groups. We verify this by confirming that the standardized mean differences (SMDs) for all pre-reform covariates are reduced to below the conventional 0.1 threshold

after matching, as presented in Section 4.1. Second, the TWFE model relies on the parallel trends assumption, which posits that the health outcomes for both groups would have followed parallel paths in the absence of the 2015 reform. We validate this assumption in Section 4.1 and 4.2, both visually and formally by testing whether the event study coefficients for the pre-treatment period are statistically indistinguishable from zero.

All statistical analyses were conducted using Python version 3.12. Statistical significance was defined at the $p < 0.1$, $p < 0.05$, and $p < 0.01$ levels, as indicated in the tables.

### 3.2. Variable setting

The variables used in the analysis are as follows. The dependent variables were the mental and physical health status of household heads. Mental health was measured by the level of depression in household heads using the CESD-11 scale, based on the CES-D (Center for Epidemiologic Studies-Depression Scale) developed by [43] and revised into the Korean version [44]. The scale includes items such as 'poor appetite', 'feeling depressed', 'just as good as other people', 'feeling that everything was an effort', 'feeling lonely', 'restless sleep', 'living without complaint', 'feeling that people were unfriendly', 'feeling that people dislike me', 'feeling sad', and 'could not get going', with each item rated on a 4-point Likert scale (1 = Rarely – 4 = All of the time). The variable for mental health is the sum of the scores for each item on this scale. Physical health was measured by the health status of residents, rated on a 5-point Likert scale (1 = Very poor – 5 = Very good).

The key independent variables for our event study design are a series of interaction terms. These are constructed by interacting a treatment group indicator (equal to 1 for households that ever receive the allowance post-2015, and 0 for the matched control group) with a set of dummy variables for each year relative to the 2015 reform. This specification allows us to estimate the dynamic, year-by-year treatment effects, while also testing the parallel pre-trends assumption.

A rich set of control variables were used for two primary purposes. First, pre-reform demographic and household characteristics were used in the propensity score matching model to construct a balanced control group and minimize selection bias. Second, a set of time-varying covariates were included in the final two-way fixed effects regression model to improve the statistical precision of the event study estimates. These include factors known to affect health, such as chronic disease, disability status, smoking, and alcohol dependence [45–48]. To avoid potential post-treatment bias, our selection of these time-varying covariates is limited to variables less likely to be immediately influenced by the housing allowance. Furthermore, we confirmed in a sensitivity analysis that our main findings are robust to the exclusion of these time-varying controls.

Gender, married, chronic disease, disabled household, smoking, area of residence, housing ownership, and alcohol dependence were measured as dummy variables. The area of residence was divided into capital and non-capital areas. Age and floor area were measured as continuous variables. Alcohol dependence was measured using the Alcohol Use Disorders Identification Test (AUDIT). The AUDIT, developed by the World Health Organization, assesses harmful drinking behaviour, alcohol dependence, and risky drinking [49]. This study classified residents with a score exceeding 20 points as alcohol abusers.

## 4. Results

### 4.1. Descriptive statistics and covariate balance

To address potential selection bias, we first performed propensity score matching. The initial, unmatched sample for matching consisted of 302 treated households and 3,184 potential control households. After performing one-to-one matching, we obtained a final analytical sample of 278 treated households and 278 matched control households. The matching process successfully retained 92.1% of the original treated group, indicating a high-quality match and preserving the representativeness of the sample.

Table 1 presents the detailed descriptive statistics and covariate balance for these groups. The pre-matching data reveals that the two groups were vastly different, with severe imbalance across most covariates such as Marital Status

**Table 1. Descriptive statistics and covariate balance before and after PSM.**

| Variables | Before Matching | | | After Matching | | |
|---|---|---|---|---|---|---|
| | Control group (N = 3,184) | Treated group (N = 302) | SMD | Control group (N = 278) | Treated group (N = 278) | SMD |
| Mental health | 3.70 | 9.81 | 1.016 | 6.87 | 9.75 | 0.412 |
| Physical health | 3.56 | 2.44 | −1.180 | 2.98 | 2.46 | −0.537 |
| Gender (1 = female) | 4.6% | 22.2% | 0.534 | 22.3% | 20.9% | −0.035 |
| Age | 67.34 | 74.44 | 0.493 | 76.21 | 74.30 | −0.130 |
| Married (1 = currently married) | 82.7% | 42.4% | −0.917 | 42.4% | 45.7% | 0.065 |
| Chronic disease (1 = yes) | 45.4% | 81.8% | 0.818 | 82.0% | 80.2% | −0.046 |
| Disabled household (1 = yes) | 11.1% | 33.8% | 0.563 | 30.9% | 32.0% | 0.023 |
| Smoking (1 = yes) | 54.9% | 59.3% | 0.088 | 56.5% | 58.3% | 0.036 |
| Alcohol dependence (1 = alcohol abuse) | 4.8% | 5.3% | 0.024 | 7.6% | 5.4% | −0.088 |
| Area of residence (1 = capital area) | 38.6% | 29.8% | −0.187 | 27.0% | 30.6% | 0.080 |
| Housing ownership (1 = owner-occupied) | 61.8% | 27.5% | −0.736 | 34.2% | 29.5% | −0.101 |
| Floor area | 79.15 | 49.31 | −1.047 | 53.35 | 51.23 | −0.087 |

(SMD = −0.917) and Floor Area (SMD = −1.047). This confirms the presence of significant selection bias in the original sample. After matching, the balance between the two groups on the baseline covariates improved dramatically. As shown in the "After Matching" panel of Table 1, the standardized mean differences for almost all covariates were reduced to well below the 0.1 threshold.

Fig 1 visually confirms this result, showing that the standardized mean differences for all covariates are clustered tightly around zero after matching. This successful balancing of key confounding variables provides a robust foundation for the subsequent causal analysis. (Fig 1).

Fig 2 presents the yearly mean trends of the two primary health outcomes—CESD score and health status—for the treatment and control groups. The primary purpose of this visualization is to inspect the parallel trends assumption prior to the 2015 policy reform.

In the pre-treatment period (2009–2014), the trends for both groups move in a remarkably parallel fashion for both outcomes. In the left panel (CESD Score), while the treatment group consistently reports higher depression scores, the year-to-year fluctuations closely track those of the control group. Similarly, in the right panel (Health Status), the two groups' trends mirror each other, maintaining a stable gap before the policy intervention. This strong visual evidence supports the validity of the parallel trends assumption, a key requirement for our difference-in-differences design.

In the post-treatment period (2015 onwards), the patterns appear to shift. For the CESD score, the gap between the two groups narrows in the years following the reform, suggesting a potential reduction in depression for the treatment group relative to the control group. For health status, a clear visual divergence is less apparent, as the groups continue to trend together, underscoring the need for a formal regression analysis to estimate the net policy effect. The following section presents the results of this formal event study analysis. (Fig 2).

## 4.2. Dynamic effects of the housing allowance

The primary results of our two-way fixed effects (TWFE) event study model are presented in Fig 3 and Table 2. Fig 3 visualizes the year-by-year estimates of the program's impact, while Table 2 provides the detailed regression output, including the coefficients for all control variables.

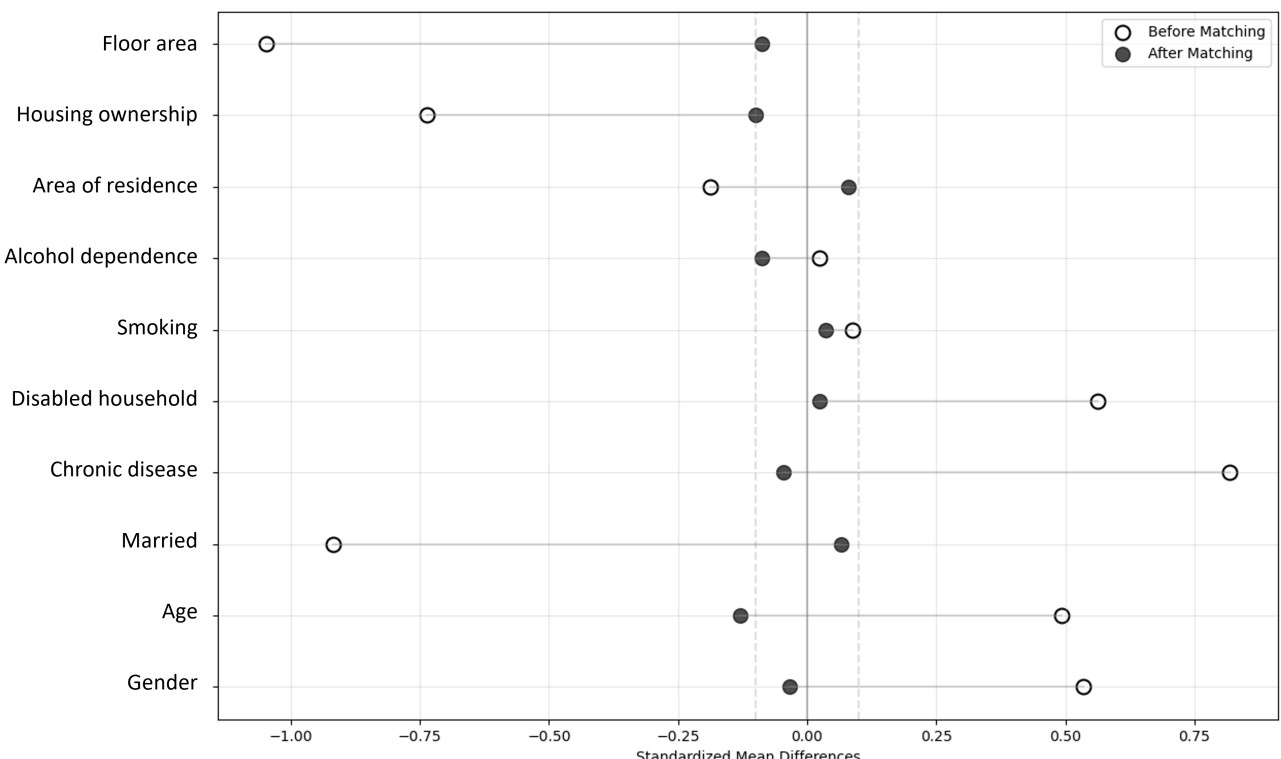

**Fig 1. Covariate balance before and after propensity score matching.** This Fig visually confirms that the standardized mean differences for all covariates are clustered tightly around zero after matching. This successful balancing of key confounding variables provides a robust foundation for the subsequent causal analysis.

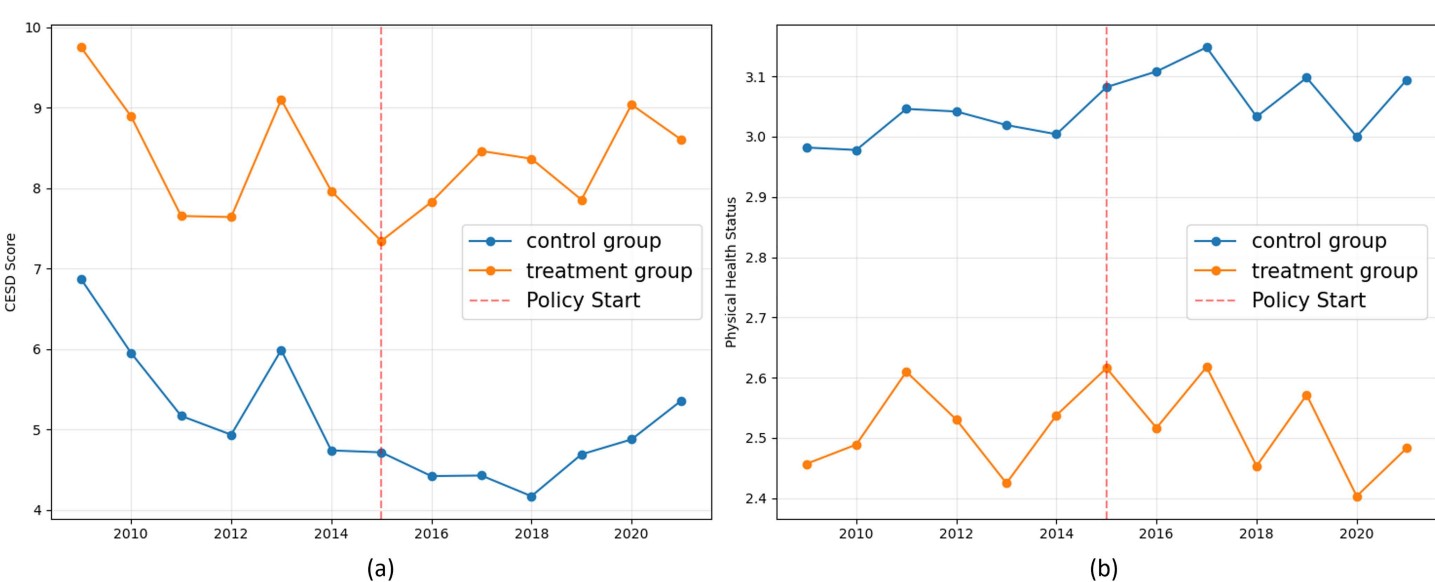

**Fig 2. Trends in mean outcomes for CESD score (a) and physical health status (b).** Yearly mean trends for the treatment and control groups are compared from 2009 to 2021. The red vertical dashed line marks the start of the policy in 2015. (a) Trends in CESD score; (b) Trends in physical health status.

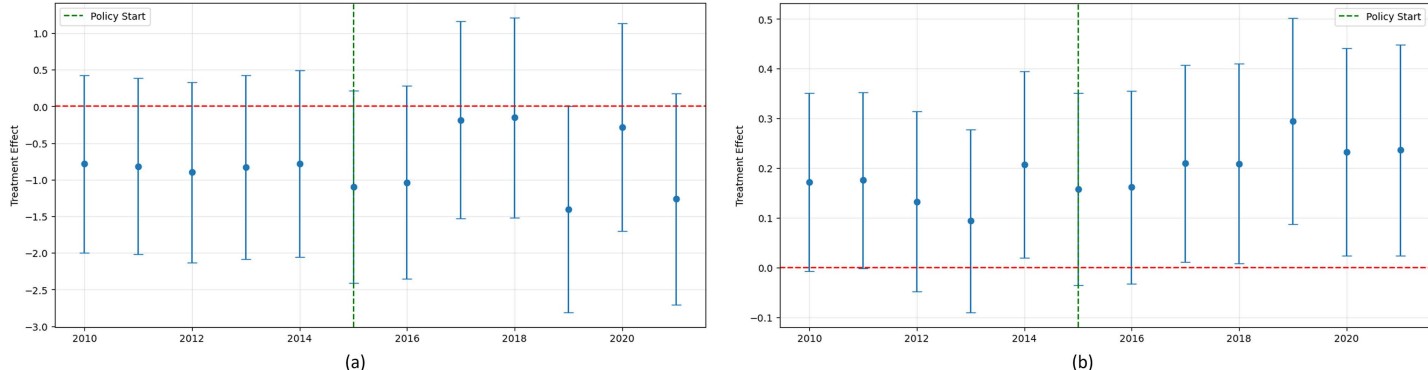

**Fig 3. Event study of the impact of housing allowance on CESD score (a) and physical health status (b).** Year-by-year estimates of the policy's impact are presented with 95% confidence intervals (error bars). The vertical green dashed line marks the 2015 policy start. (a) Impact on CESD score; (b) Impact on physical health status.

A critical component of this analysis is the validation of the parallel trends assumption. The coefficients for the pre-treatment period (2010–2014) in Table 2 serve this purpose. For the CESD Score model, none of the pre-treatment coefficients are statistically significant, providing strong evidence that the parallel trends assumption holds. For the Health Status model, a statistically significant difference appears in 2014 ($\beta = 0.208$, $p < 0.05$), though the coefficients for all other pre-treatment years are insignificant. Given that the majority of the pre-treatment coefficients are statistically indistinguishable from zero, and the visual evidence from Fig 2 suggests parallel pre-trends, we conclude that our research design is valid. (Fig 3).

The post-treatment coefficients in Table 2 reveal the dynamic effects of the policy. For mental health, the results show a delayed positive effect. While the impact in the immediate post-reform years is not statistically significant, a significant reduction in depression emerges four years after the reform, in 2019 ($\beta = -1.401$, $p < 0.1$) and again in 2021 ($\beta = -1.264$, $p < 0.1$). This suggests that the mental health benefits of the housing allowance may take several years to fully materialize.

For physical health, the program's positive impact appears more quickly and is sustained. A statistically significant improvement in health status begins to emerge two years post-reform in 2017 ($\beta = 0.210$, $p < 0.05$) and continues through 2021. The effect is largest in 2019 ($\beta = 0.295$, $p < 0.01$), indicating that the benefits of the housing allowance on physical health are substantial and accumulate over time.

The results for the control variables are also largely consistent with theoretical expectations. The main effect for the Treatment Group indicates that, on average, the recipient group had significantly higher depression ($\beta = 0.965$, $p < 0.1$) and poorer physical health ($\beta = -0.354$, $p < 0.01$) than the control group, confirming their underlying vulnerability. This finding aligns with previous research indicating that the residential environment is significantly related to the level of depression among residents [50]. For both health outcomes, Age and Chronic Disease were highly significant predictors in the expected directions. In the mental health model, being married ($\beta = -1.180$, $p < 0.01$) and having a larger living space ($\beta = -0.014$, $p < 0.01$) were associated with lower depression scores, while alcohol dependence was associated with higher scores ($\beta = 1.358$, $p < 0.01$). In the physical health model, the presence of a disability was a strong predictor of poorer health status ($\beta = -0.315$, $p < 0.01$). These findings enhance the overall validity of our model specification.

## 5. Discussion and conclusion

The residential environment is closely linked to residents' health, with poor conditions adversely affecting well-being. The South Korean government has introduced various housing assistance programmes to alleviate housing cost burdens, and recently, it has implemented demand-oriented housing policies by expanding the benefits and eligibility of the housing

**Table 2. Analysis of the impact of housing allowance on mental health.**

| Variables | | Mental health (CESD Score) | | physical health (Health status) | |
|---|---|---|---|---|---|
| | | Coef. | S.E. | Coef. | S.E. |
| **Event Study Coefficients** | | | | | |
| **Pre-Treatment Trends** | | | | | |
| Year = 2010 (t-5) | | −0.785 | −0.62 | 0.172* | −0.091 |
| Year = 2011 (t-4) | | −0.818 | −0.613 | 0.176* | −0.09 |
| Year = 2012 (t-3) | | −0.901 | −0.627 | 0.133 | −0.092 |
| Year = 2013 (t-2) | | −0.829 | −0.638 | 0.094 | −0.094 |
| Year = 2014 (t-1) | | −0.778 | −0.651 | 0.208** | −0.096 |
| **Post-Treatment Effects** | | | | | |
| Year = 2015 (t = 0) | | −1.095 | −0.669 | 0.158 | −0.099 |
| Year = 2016 (t + 1) | | −1.038 | −0.672 | 0.162 | −0.099 |
| Year = 2017 (t + 2) | | −0.183 | −0.687 | 0.210** | −0.101 |
| Year = 2018 (t + 3) | | −0.151 | −0.696 | 0.209** | −0.103 |
| Year = 2019 (t + 4) | | −1.401* | −0.719 | 0.295*** | −0.106 |
| Year = 2020 (t + 5) | | −0.283 | −0.724 | 0.233** | −0.107 |
| Year = 2021 (t + 6) | | −1.264* | −0.737 | 0.236** | −0.108 |
| **Control Variables** | | | | | |
| Treatment Group | | 0.965* | −0.506 | −0.354*** | −0.075 |
| Gender (1 = female) | | 0.62 | −0.444 | 0.073 | −0.065 |
| Age | | 0.122*** | −0.018 | −0.029*** | −0.003 |
| Married (1 = currently married) | | −1.180*** | −0.389 | 0.088 | −0.057 |
| Chronic disease (1 = yes) | | 0.474** | −0.239 | −0.390*** | −0.035 |
| Disabled household (1 = yes) | | 0.274 | −0.422 | −0.315*** | −0.062 |
| Smoking (1 = yes) | | −0.217 | −0.243 | 0.05 | −0.036 |
| Alcohol dependence (1 = alcohol abuse) | | 1.358*** | −0.428 | −0.001 | −0.063 |
| Area of residence (1 = capital area) | | 1.09 | −0.838 | 0.055 | −0.123 |
| Housing ownership (1 = owner-occupied) | | −0.37 | −0.292 | 0.041 | −0.043 |
| Floor area | | −0.014*** | −0.005 | 0.001** | −0.001 |
| Model statistics | Number of Observations | 4,951 | | | |
| | Number of Households | 556 | | | |
| | Household Fixed Effects | Yes | | Yes | |
| | Year Fixed Effects | Yes | | Yes | |
| | R-squared (Within) | 0.030 | | 0.075 | |
| | Adjusted R-squared (Within) | 0.026 | | 0.071 | |
| | RMSE | 4.479 | | 0.659 | |
| | F-Statistic | 5.761 | | 17.490 | |

***$p < 0.01$, **$p < 0.05$, *$p < 0.1$

allowance system. It is necessary to verify whether these housing policies reduce the housing cost burden for recipients and positively impact their health.

This study analysed the impact of the housing allowance system on household heads' mental and physical health using data from the Korea Welfare Panel Study (2009–2021). To provide a rigorous causal assessment, this study employed

propensity score matching (PSM) to construct a comparable control group, followed by a two-way fixed effects (TWFE) event study model to analyse the dynamic effects of the 2015 policy expansion. The analysis revealed that the program's positive health effects were not immediate but emerged over time. For mental health, a statistically significant reduction in depression was observed approximately four years after the reform, in 2019 and 2021. Similarly, the program's positive impact on physical health was also delayed, with a consistent and statistically significant improvement in health status emerging from 2017 onwards. These findings indicate that the housing allowance policy has a positive, albeit delayed, effect on both the mental and physical health of recipients.

Our findings that South Korea's housing allowance program improves both mental and physical health are consistent with a growing body of international literature demonstrating the health benefits of rental assistance. For example, our result showing a reduction in depression aligns with findings that rental assistance recipients report significantly less psychological distress [5]. Similarly, our finding of improved physical health is supported by U.S. studies linking housing assistance to better management of chronic conditions, such as a reduced likelihood of uncontrolled diabetes [35]. Furthermore, the mechanisms for these health improvements may be similar across contexts. A key pathway identified in the literature is the alleviation of material hardship, which lessens the difficult trade-offs between housing and other necessities. By situating our findings within this extensive body of work, our study suggests that demand-side housing policies, like South Korea's allowance program, can be an effective public health tool.

This study has several important implications. Our findings confirm that the housing allowance policy positively affects mental and physical health, underscoring the importance of housing welfare as a critical component of social policy. Investing in housing welfare programmes can enhance overall well-being and reduce health disparities. However, our event study analysis reveals a crucial nuance: these health benefits are not immediate and take several years to become statistically significant. This finding suggests that the full returns of housing allowances should be viewed as a long-term investment in public health, and that short-term evaluations may understate their true value. Furthermore, this highlights the need for more precise identification of policy beneficiaries and tailored, sustained support based on individual characteristics and needs.

However, this study has limitations. First, our analysis is confined to the health of household heads, which may not capture the full impact of the program on the entire household. The effects of the housing allowance could differ for other members, such as spouses or children, whose health trajectories may respond differently to changes in housing stability and financial resources. Second, the study did not account for changes in the broader environment, such as shifts in policy or economic conditions during the study period. Factors like economic crises, real estate market fluctuations, and changes in employment rates may influence the relationship between housing allowance and health, potentially distorting the policy effects. Additionally, the study simplified the impact of the housing allowance programme on health, overlooking the complex factors related to overall quality of life. Future research should examine the health of all household members and incorporate a wider range of quality-of-life factors to provide more comprehensive and reliable analysis results.

## Author contributions

**Conceptualization:** Saehim Kim.

**Data curation:** Saehim Kim, Saebae Ryu, Kyuhyun Park.

**Formal analysis:** Saebae Ryu, Kyuhyun Park.

**Investigation:** Saehim Kim.

**Methodology:** Saehim Kim, Saebae Ryu.

**Project administration:** Saehim Kim.

**Supervision:** Myeong-Hun Lee.

**Validation:** Saehim Kim, Saebae Ryu.

**Visualization:** Saebae Ryu.

**Writing – original draft:** Saehim Kim.

**Writing – review & editing:** Saehim Kim.

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
