## [Decision Letter · Decision Letter 0]

13 Jul 2025

PONE-D-25-23017Impact of housing allowance programme on the physical and mental health of households in South KoreaPLOS ONE

Dear Dr. Lee,

Thank you for submitting your manuscript to PLOS ONE. After careful consideration, we feel that it has merit but does not fully meet PLOS ONE’s publication criteria as it currently stands. Therefore, we invite you to submit a revised version of the manuscript that addresses the points raised during the review process.

The manuscript has been evaluated by two reviewers, and their comments are available below.

The reviewers have raised a number of major concerns. They request improvements to the reporting of methodological aspects of the study, for example, regarding the exclusion criteria and more information on how the data collection was completed, as well as greater expansion in the discussion section.

Could you please carefully revise the manuscript to address all comments raised?

We look forward to receiving your revised manuscript.

Kind regards,

Avanti Dey, PhD

Staff Editor

PLOS ONE

Journal Requirements:

Reviewers' comments:

Reviewer's Responses to Questions

**Comments to the Author**

1. Is the manuscript technically sound, and do the data support the conclusions?

Reviewer #1: No

Reviewer #2: Yes

2. Has the statistical analysis been performed appropriately and rigorously? 

Reviewer #1: No

Reviewer #2: Yes

3. Have the authors made all data underlying the findings in their manuscript fully available?

Reviewer #1: Yes

Reviewer #2: Yes

4. Is the manuscript presented in an intelligible fashion and written in standard English?

Reviewer #1: Yes

Reviewer #2: Yes

5. Review Comments to the Author

Reviewer #1: This manuscript examines the effect of a housing allowance program in South Korea on reported physical and mental health outcomes of renters. The authors use a panel survey and difference in difference methods to examine the effects of the expansion of the program on reported outcomes, demonstrating that program recipients experienced health improvements following expansion. My main concerns relate to the lack of detail surrounding the program and the expansion and to the interpretation of the DID results. I offer suggestions to the authors to improve their work.

Major concerns:

More detail should be provided up front about what the housing allowance program actually does. For readers outside South Korea, there may not be familiarity with this type of program. Although the authors describe the program in the lit review section, more could be said in the introduction to motivate the study. It is not clear until page 5 whether this program is similar to the US housing voucher program or whether the subsidy is more open-ended. And even in this paragraph, the description of what the program actually provides is somewhat vague. Finally, even after getting to the end of this section, it was not entirely clear whether the housing allowance was provided as a cash transfer or in the form of a voucher. Finally, more detail should be provided on exactly how the expansion of the program worked in terms of the magnitude of what it provided.

There should be a more thorough justification of the inclusion of some characteristics in the model. The DID model should difference out time-invariant differences between the treatment and control groups, so there should not be a need to control for these. If they are included to enhance statistical precision, this should be made clear. As for time-varying covariates that differ between treatment and control, more attention should be paid to why we would expect these not to be on the pathway from the policy to the outcome. Are some of the covariates included (e.g. smoking status) time varying? If so, this could create serious bias if treatment affects these variables.

It is not clear to me who the control group is. Is it just households not receiving the housing allowance? How is this determined? Why would we expect trends in this group to be similar to those of the treatment group?

Similarly, is receiving the housing allowance a time-varying variable? If so, are some respondents receiving the allowance after 2015 but not before?

At the end of the first paragraph of the Data Analysis and Methodology section, there is a sentence that mentions that data deemed outliers were excluded, but there is no more detail about how this was determined. Outliers on which characteristics? How many were outliers? Just because an individual is an outlier, does not mean they should be excluded. Much more detail is needed.

DID assumes that—in the absence of treatment—trends for the treatment and control group would be identical. The parallel trends assumption cannot be tested, but must be justified based on theory and supporting empirical results. Authors often show event study results to confirm that treatment effects indeed only show up after the treatment. Something like this would go a long way to assuage my concerns that the models are simply picking up unrelated background trends.

Minor concerns

In Table 6, the coefficient for the DID variable is 0.08 and the standard error is 0.02, but it is not listed as significant.

No attempt is made in the results section to interpret the effect size. All statements are directional, but it is not clear if the effect is meaningfully large or reasonably sized.

DID models should also include dummy variables for individual cases. Although those variables may be in the model, it is not clear from Tables 5 and 6 whether this is the case.

Reviewer #2: Literature review:

-Consider citing "Bhat, A. C., Fenelon, A., & Almeida, D. M. (2025). Housing insecurity pathways to physiological and epigenetic manifestations of health among aging adults: a conceptual model. Frontiers in Public Health, 13, 1485371." to provide a conceptual model basis for how housing assistance programs can ameliorate the adverse health impacts of housing insecurity

-Please add more literature to support the disproportionate impacts on women/ethnic minorities/older adults. What are the factors that contribute to vulnerabilities for these groups? Also explain how this ties into the analyses you are conducting in the context of Korea.

Discussion:

-Both discussion and review of literature sections are a bit thin. There are several more studies than what you cited that discuss how governmental assistance programs ameliorate housing insecurity and related health effects (physical and mental health). Look into more of the work by Andrew Fenelon and others who have published extensively in this area and incorporate into these sections.

-Go more into depth regarding the similarities and differences in housing assistance programs in countries such as the US/UK and Korea

-In the US access to housing assistance programs also increases access to health insurance (Simon, A. E., Fenelon, A., Helms, V., Lloyd, P. C., & Rossen, L. M. (2017). HUD housing assistance associated with lower uninsurance rates and unmet medical need. Health Affairs, 36(6), 1016-1023.). Would this be expected to translate similarly in the context of Korea?

-Consider whether only examining household heads' health is a limiting factor, rather than examining health of multiple members of the household. Write about this in limitations.

6. PLOS authors have the option to publish the peer review history of their article (what does this mean? ). If published, this will include your full peer review and any attached files.

**Do you want your identity to be public for this peer review?** For information about this choice, including consent withdrawal, please see our Privacy Policy .

Reviewer #1: No

Reviewer #2: No

---

## [Author Response · Author response to Decision Letter 1]

4 Sep 2025

Thank you for your valuable and constructive feedback. We have thoroughly addressed all comments in the attached 'Response to Reviewers' file. The manuscript has been substantially revised in accordance with your suggestions, particularly regarding the analytical methodology and the expansion of the literature review.

---

## [Decision Letter · Decision Letter 1]

30 Oct 2025

PONE-D-25-23017R1Impact of housing allowance programme on the physical and mental health of households in South KoreaPLOS One

Dear Dr. Lee,

Thank you for submitting your manuscript to PLOS ONE. After careful consideration, we feel that it has merit but does not fully meet PLOS ONE’s publication criteria as it currently stands. Therefore, we invite you to submit a revised version of the manuscript that addresses the points raised during the review process.

The manuscript has been evaluated by two reviewers, and their comments are available below. Could you please revise the manuscript to carefully address the concerns raised?

We look forward to receiving your revised manuscript.

Sincerely,

Alejandro Torrado Pacheco, PhD

Associate Editor

PLOS One

Journal Requirements:

Reviewers' comments:

Reviewer's Responses to Questions

**Comments to the Author**

1. If the authors have adequately addressed your comments raised in a previous round of review and you feel that this manuscript is now acceptable for publication, you may indicate that here to bypass the “Comments to the Author” section, enter your conflict of interest statement in the “Confidential to Editor” section, and submit your "Accept" recommendation.

Reviewer #2: All comments have been addressed

Reviewer #3: (No Response)

2. Is the manuscript technically sound, and do the data support the conclusions?

Reviewer #2: Yes

Reviewer #3: Yes

3. Has the statistical analysis been performed appropriately and rigorously? 

Reviewer #2: Yes

Reviewer #3: Yes

4. Have the authors made all data underlying the findings in their manuscript fully available?

Reviewer #2: Yes

Reviewer #3: Yes

5. Is the manuscript presented in an intelligible fashion and written in standard English?

Reviewer #2: Yes

Reviewer #3: Yes

6. Review Comments to the Author

Reviewer #2: Thank you for addressing reviewer comments. The manuscript is vastly improved now that authors have addressed all reviewer comments.

Reviewer #3: An interesting study with comprehensive analyses.

The manuscript could be further improved.

Page 12 Line 2-4: The sentence could be improved e.g. A TWFE event study replaces a single pre-post treatment indicator with a series of year-by-year interaction terms. This approach allows researchers to formally test the key parallel trends assumption and to trace the dynamic, year-over-year effects of the policy following the 2015 reform (Clarke 5 & Tapia-Schythe, 2021; Sun & Abraham, 2021).

The statistical software used, including its publisher and version, along with the level of significant applied are to be stated.

Table 2: Adjusted R^2, RMSE, F-Statistic could be presented.

The assumptions fulfilment for the statistical test are to be mentioned.

Table 1, 2: The N for the intervention group and control group are to be stated.

All references in the text and the reference list are to follow journal’s format.

7. PLOS authors have the option to publish the peer review history of their article (what does this mean? ). If published, this will include your full peer review and any attached files.

**Do you want your identity to be public for this peer review?** For information about this choice, including consent withdrawal, please see our Privacy Policy .

Reviewer #2: **Yes:** Aarti C. Bhat

Reviewer #3: No

---

## [Author Response · Author response to Decision Letter 2]

2 Nov 2025

Thank you for your valuable feedback. We have addressed all comments in detail in our rebuttal letter, which has been uploaded as a separate file labeled 'Response to Reviewers'.

---

## [Decision Letter · Decision Letter 2]

13 Feb 2026

Impact of housing allowance programme on the physical and mental health of households in South Korea

PONE-D-25-23017R2

Dear Dr. Lee,

We’re pleased to inform you that your manuscript has been judged scientifically suitable for publication and will be formally accepted for publication once it meets all outstanding technical requirements.

Kind regards,

Marianne Clemence

Staff Editor

PLOS One

Additional Editor Comments (optional):

Reviewers' comments:

Reviewer's Responses to Questions

**Comments to the Author**

1. If the authors have adequately addressed your comments raised in a previous round of review and you feel that this manuscript is now acceptable for publication, you may indicate that here to bypass the “Comments to the Author” section, enter your conflict of interest statement in the “Confidential to Editor” section, and submit your "Accept" recommendation.

Reviewer #3: All comments have been addressed

2. Is the manuscript technically sound, and do the data support the conclusions?

Reviewer #3: (No Response)

3. Has the statistical analysis been performed appropriately and rigorously? 

Reviewer #3: (No Response)

4. Have the authors made all data underlying the findings in their manuscript fully available?

Reviewer #3: (No Response)

5. Is the manuscript presented in an intelligible fashion and written in standard English?

Reviewer #3: (No Response)

6. Review Comments to the Author

Reviewer #3: (No Response)

7. PLOS authors have the option to publish the peer review history of their article (what does this mean? ). If published, this will include your full peer review and any attached files.

**Do you want your identity to be public for this peer review?** For information about this choice, including consent withdrawal, please see our Privacy Policy .

Reviewer #3: No

---

## [Editor Report · Acceptance letter]

PONE-D-25-23017R2

PLOS One

Dear Dr. Lee,

I'm pleased to inform you that your manuscript has been deemed suitable for publication in PLOS One. Congratulations! Your manuscript is now being handed over to our production team.

Kind regards,

on behalf of

Dr Marianne Clemence

Staff Editor

PLOS One